# Oxidative Stress as a Reliable Biomarker of Carotid Plaque Instability: A Pilot Study

**DOI:** 10.3390/antiox12020506

**Published:** 2023-02-17

**Authors:** Norbert Svoboda, Karolina Kočí, Anna Seidlová, Václav Mandys, Jiří Suttnar, Alžběta Hlaváčková, Ondřej Kučerka, David Netuka, Martin Malý

**Affiliations:** 1First Faculty of Medicine, Charles University, 169 02 Prague, Czech Republic; 2University Military Hospital Prague, 168 02 Prague, Czech Republic; 3Third Faculty of Medicine, Charles University, 100 00 Prague, Czech Republic; 4Institute of Haematology and Blood Transfusion, 120 00 Prague, Czech Republic; 5Faculty of Military Health Sciences, University of Defence, 500 02 Hradec Kralove, Czech Republic

**Keywords:** carotid stenosis, plaque, atherosclerotic, oxidative stress, endarterectomy, carotid, histology

## Abstract

**Background:** Predicting stroke risk in patients with carotid artery stenosis (CS) remains challenging. Circulating biomarkers seem to provide improvements with respect to risk stratification. **Methods:** Study patients who underwent carotid endarterectomy were categorized into four groups according to symptomatology and compared as follows: symptomatic with asymptomatic patients; and asymptomatic patients including amaurosis fugax (AF) (asymptomatic + AF group) with patients with a transient ischemic attack (TIA) or brain stroke (BS) (hemispheric brain stroke group). Carotid specimens were histologically analyzed and classified based on the American Heart Classification (AHA) standard. As a marker of OS, the plasma levels of malondialdehyde (MDA) were measured. Comparisons of MDA plasma levels between groups were analyzed. **Results**: In total, 35 patients were included in the study. There were 22 (63%) patients in the asymptomatic group and 13 (37%) in the symptomatic group. Atheromatous plaque (*p =* 0.03) and old hemorrhage (*p =* 0.05), fibrous plaque (*p =* 0.04), myxoid changes (*p =* 0.02), plaques without hemorrhage (*p =* 0.04), significant neovascularization (*p =* 0.04) and AHA classification (*p =* 0.006) had significant correlations with clinical presentation. There were 26 (74%) patients in the asymptomatic group and 9 (26%) in the hemispheric brain stroke group. Atheromatous plaque (*p =* 0.02), old hemorrhage (*p =* 0.05) and plaques without neovascularization (*p =* 0.02), fibrous plaque (*p =* 0.03), plaques without hemorrhage (*p =* 0.02) and AHA classification (*p =* 0.01) had significant correlations with clinical presentation. There was no significant difference between symptomatic and asymptomatic groups with respect to MDA plasma levels (*p* = 0.232). A significant difference was observed when MDA plasma levels were compared to asymptomatic + AF and the hemispheric stroke group (*p* = 0.002). **Conclusions**: MDA plasma level correlates with the risk of hemispheric stroke (TIA or BS) and is a reliable marker of plaque vulnerability in carotid artery stenosis.

## 1. Introduction

Despite progressive and extensive research, the exact link between atherosclerotic carotid stenosis (CS) and brain stroke remains unclear. CS causes up to 15% of all ischemic brain strokes [1]. However, not every CS places patients at the same risk of brain stroke [2]. The former hypothesis, i.e., that the risk of brain stroke is dominantly dependent on the severity of stenosis, is no longer considered valid [2]. It seems that the risk of brain stroke in patients with CS depends on several factors, such as the severity of stenosis and the histological and radiological appearance of the plaque [2]. It is essential to select patients who have high-risk CS in order to provide them with adequate treatment. Proper selection criteria, however, have not been definitively set so far.

Previously, based on histological and radiological appearance, carotid plaques were divided into stable and unstable plaques [3,4]. Patients with stable atherosclerotic plaques have a low risk of brain stroke, whereas unstable plaques are prone to embolization, leading to brain stroke. Although multiple studies have analyzed the histological structure of carotid plaques, the impact of a single histological feature on plaque stability remains elusive [5,6,7]. On the other hand, it seems that the appearance of complex plaques is an important prognostic factor [8,9]. However, at present, predicting the risk of stroke in patients with CS remains challenging.

Therefore, other predictive markers are required. A recently introduced analysis of circulating biomarkers seems to be a promising method for detecting vulnerable CS [2]. One reasonable biomarker seems to be the factors influencing oxidative stress (OS) [10]. OS is characterized by the increased production of reactive oxygen species (ROSs) and reactive nitrogen species (RNSs) [11], which are not sufficiently balanced by antioxidant defenses due to the disruption of redox signaling and adaptation. Knowledge of significant OS in strokes is important, given its atherosclerotic or cardioembolic etiology [12,13,14,15]. OS promotes endothelial dysfunction, platelet activation, platelet–leukocyte aggregation and thrombus formation by the modification of fibrinogen [16,17,18]. ROSs include, for example, superoxide anion radical (O_2_^•−^), hydrogen peroxide, peroxyl radicals and hydroxyl radicals [19]. RNSs include nitric oxide (^•^NO) and peroxynitrite anion (ONOO^−^). The enzymatic oxidation of arginine produces NO by three isoforms of NO synthases: inducible NO synthase (iNOS), endothelial NO synthase (eNOS) and neuronal NO synthase (nNOS). Isoforms iNOS and nNOS are harmful in ischemic stroke [20]. The presence of OS can be monitored using different methods, mainly LC-MS/MS, to measure carbonylated proteins, MDA, 4-hydroxy-2-nonenal and F2-isoprostanes [21]. MDA, belonging to the family of lipid peroxidation biomarkers, is considered a standard and reliable OS marker [22]. These biomarkers are also important triggers of stroke-related thrombosis [23]. The correlation between MDA plasma levels and CS vulnerability is unknown.

This study aims to analyze carotid plaque vulnerability concerning its histological structure and the influence of OS on carotid plaque vulnerability by measuring the plasma levels of malondialdehyde (MDA).

## 2. Materials and Methods

### 2.1. Study Design

This is a single center observational case-control study. Because of the nature of an observational study, inclusion/exclusion criteria strictly followed the current guidelines for carotid stenosis surgery [24]. Every patient from whom MDA plasma levels were obtained and who underwent carotid endarterectomy for CS within the period from 2019 and 2020 was included in the study. Informed consent was obtained from all subjects involved in the study. The study was approved by the Ethics Committee of the Central Military Hospital Prague.

Patients were divided into four groups based on two different classifications of symptomatology. Firstly, groups were defined with respect to the North American Symptomatic Carotid Endarterectomy Trial (NASCET) [25] and European Carotid Surgery Trial (ECST) [26]. The trial defined symptomatic patients as those who presented either with a brain stroke (BS), transient ischemic attack (TIA) or amaurosis fugax (AF) within the previous 6 months before surgery (symptomatic group). Patients with no symptoms or with symptoms that occurred more than 6 months before the surgery were classified as asymptomatic (asymptomatic group). Secondly, based on knowledge from histological analyses that the carotid plaques of patients with AF have similar features as asymptomatic plaques [6], patients were divided into a group of asymptomatic patients, i.e., those with AF (asymptomatic + AF group), and into a group of patients who presented hemispheric brain strokes, namely, BS or TIA (hemispheric brain stroke group).

Each patient underwent carotid plaque assessment by ultrasound (US), followed by computed tomography angiography to corroborate the US diagnosis.

The indication criteria for carotid artery stenosis surgery strictly followed current guidelines [24]. During surgery, surgeons sought to remove the atherosclerotic plaque in one piece to avoid rupturing the plaque wall. Damaged plaques were excluded from the study.

### 2.2. Histochemical Analysis

Immediately after removal, harvested endarterectomy specimens were placed in 10% formaldehyde. Representative parts of the specimen were cross-sectioned in approximately 4 mm thick samples. In further processing, the samples were decalcified by a hydrochloric acid solution and embedded in paraffin. The samples were cut into five-micron-thick tissue sections. Xylene was used as a deparaffinization agent, and graded alcohol was used for the hybridization of the tissue sections. For staining parallel sections, hematoxylin and eosin with the van Gieson/orcein method were used. The indirect immunohistologic method was used for the detection of endothelial cells (CD31 marker, primary mouse anti-human monoclonal antibody and clone JC70A) and macrophages (CD68 marker, primary mouse anti-human monoclonal antibody and clone PG-M1). All histological analyses were performed by one experienced pathologist (VM) using a bright-field optical microscope (Nikon Eclipse E 400).

Endarterectomy specimens were scanned for multiple histological features, including eccentricity, the presence of atheromatous or fibrous tissue, calcification, myxoid change, hemorrhage, thrombosis, inflammation, foamy macrophage, giant cell reaction, hemosiderin, neovascularisation or ossification (TAB 1). All specimens were divided into AHA groups IV/V, VIII, or VI, according to the AHA classification [9]. Plaques in the AHA VI group were gathered in the group of unstable plaques.

### 2.3. MDA Analysis

At the time of US examination, blood samples were obtained from patients. The blood samples were kept in vacutainer tubes containing EDTA and centrifuged immediately at 4000× *g* for 5 min at 4 °C. Until the time of further laboratory analyses, plasma samples were stored in the dark at −80 °C.

The analysis of MDA was essentially performed according to Bechynska [27]. Briefly, 10 mL of diluted internal standard MDA-D2 (10 mM) was added to 100 mL of EDTA plasma and hydrolyzed with NaOH (1 M final concentration) for 30 min at 60 °C. The precipitation of protein was accomplished by adding 3 M HClO4 to the hydrolysate. Such samples were centrifuged. The supernatant was derivatized by 5 mM 2,4-dinitrophenylhydrazine (DNPH) using 30 min shaking on Vibrax in the dark at laboratory temperatures. The centrifugation of the reaction mixture was performed, and 20 mL of the mixture was injected into the HPLC column Nucleosil C18 ec (125 × 3 mm, 5 μm) (Macherey-Nagel, Düren, Germany) at 40 °C using the isocratic mobile phase comprising 0.1% of formic acid in 50% acetonitrile (*v*/*v*). The flow rate was 400 μL/min. The HPLC system was connected to mass spectrometer QTRAP 4000 (Sciex, Prague, Czech Republic). MDA and MDA-D2 DNPH derivatives (MDA-DNPH and MDA-D2-DNPH) were detected in the positive multiple reaction monitoring (MRM) mode. MDA-DNPH was monitored at *m*/*z* 235à189 and MDA-D2-DNPH at *m*/*z* 237à191. MDA and MDA-D2 DNPH derivatives were eluted at 3.00 min. Analyst v.1.6 from SCIEX was used for the acquisition and analysis of the data.

### 2.4. Statistical Analysis

Patients were grouped and compared using the following pattern: asymptomatic versus symptomatic groups and asymptomatic + AF versus hemispheric brain stroke groups. The relationship between groups and endarterectomy specimens based on individual histological features and AHA plaque classification was evaluated by the chi-square test. The Wilcoxon rank sum test with continuity correction was calculated for the MDA plasma levels of the groups. All statistical tests were performed at a significance level of 95% (*p* < 0.05). JASP 0.16.4 software was used for statistical analyses.

## 3. Results

In total, 35 patients were included in the study. Most carotid specimens were obtained from men (n = 31, 89%); other specimens were less frequently obtained from women (n = 4, 11%). The average stenosis was 76% in asymptomatic patients and 83% in symptomatic patients. The average age was 76 (55–92) years. There were 22 (63%) asymptomatic and 9 (26%) patients with BS, 4 (11%) patients with AF and no patient presented with TIA. Patient baseline characteristics are shown in Table 1.

There were 22 (63%) patients in the asymptomatic group and 13 (37%) in the symptomatic group. Several individual histological features correlated significantly: atheromatous plaque (*p* = 0.03) and old hemorrhage (*p* = 0.05) were significantly more present in the symptomatic group, whereas fibrous plaque (*p* = 0.04), myxoid changes (*p* = 0.02), plaques without hemorrhage (*p* = 0.04) and significant neovascularization (*p* = 0.04) were more common in the asymptomatic group. Plaque vulnerability based on the AHA classification had a significant correlation with clinical presentation (*p* = 0.006).

There were 26 (74%) patients in the asymptomatic + AF group and 9 (26%) in the hemispheric brain stroke group. Several individual histological features correlated significantly: atheromatous plaque (*p* = 0.02), old hemorrhage (*p* = 0.05) and plaques without neovascularization (*p* = 0.02) were found preferentially in the hemispheric brain stroke group, whereas fibrous plaque (*p* = 0.03) and plaques without hemorrhage (*p* = 0.02) were more common in the asymptomatic + AF group. Plaque vulnerability based on AHA classification had significant correlations with clinical presentation (*p* = 0.01).

Detailed analyses of individual histological features and complex plaque appearance according to the AHA classification for different groups are depicted in Table 2 and Table 3.

The comparison between asymptomatic and symptomatic groups based on MDA plasma levels did not reveal a statistically significant difference (*p* = 0.232). There was a statistically significant difference between asymptomatic + AF and hemispheric brain stroke groups based on levels of MDA in plasma (*p* = 0.002) (Figure 1).

## 4. Discussion

The search for the selection criteria of patients with CS who are in danger of brain strokes has a long history. Previously, as shown by two large DSA-based multicentric randomized studies from the early 1990s, it was believed that the risk of brain stroke from CS depends on the degree of stenosis. The analysis of endarterectomy specimens revealed that the histological structure of the plaque can more precisely predict the risk of brain stroke when combined with the severity of CS. Additionally, studying the histological appearance of endarterectomy specimens is an important approach for clarifying the exact pathophysiology of atherosclerotic CS. However, the outcomes of studies based on endarterectomy specimens are incongruent. Multiple studies have claimed that histological features such as eccentricity [28,29], calcification [30], hemorrhage [31,32] and neovascularization [7] are more commonly found in symptomatic patients. However, others presented data contradicting this view [5,6,7,11,13,33]. On the other hand, there is a general agreement that rather than individual histological features, a complex histological appearance is relevant. Hence, a complex histological plaque analysis was suggested, leading to the development of the AHA classification. The AHA classification seems to be a meaningful method for detecting plaque instability [8,9,13]. Our study found several significant individual features that correlated with patient symptomatology. However, it should be emphasized that the highest statistical correlation was observed in complex plaque characteristics delineated by the AHA classification at a *p*-value = 0.006 in asymptomatic and symptomatic groups. Interestingly, in a study by Verhoeven et al. [6], the authors analyzed 404 carotid endarterectomy specimens and compared their histological appearance regarding their symptomatology (asymptomatic, AF, TIA, stroke). It was shown that the plaques of asymptomatic patients resemble those of patients with AF. The authors concluded that the plaques of asymptomatic or AF patients have the properties of stable plaques, in contrast to the those of patients with TIA and BS. However, in our analysis, the correlation between AHA classification and the asymptomatic + AF and hemispheric brain stroke groups was lower (*p* = 0.01), although the correlation remained statistically significant. As a histological parameter, however, carotid endarterectomy specimens can only be evaluated after surgical removal, which limits the clinical impact of this procedure. With the introduction of non-invasive methods (such as ultrasound, computed tomography and magnetic resonance angiography), the presurgical appearance of the carotic plaque structure can be analyzed. It was shown that the radiological evaluation of carotid plaque composition is of great importance regarding the vulnerability of the plaque [7,13,31]. However, none of the presented radiological methods is accurate enough [2,12].

In light of the above-mentioned failures with respect to the precise evaluation of vulnerable CS, biomarkers are believed to be a reasonable path for more accurate predictions of plaque vulnerability [14,15]. The idea is based on the knowledge that changes in the plasma levels of individual biomarkers can provide information about the nature of carotid plaques. The amount of circulating biomarkers can be changed in vulnerable CS as a causal factor of atherosclerosis (lipid-related and lipoprotein biomarkers [15], MDA [16]), or it can be diffused from a carotid plaque that becomes vulnerable (apolipoprotein J, E, microRNA [15], etc.). Numerous studies have tried to find significantly specific and sensitive biomarkers. These studies differ, however, in terms of their methodologies, and this makes comparisons of individual biomarkers difficult. Up until now, there is no consensus on what an ideal biomarker predictor of CS stroke risk is. Biomarkers of oxidative stress (OS), as a cause of atherosclerosis, seem to be reasonable targets. OS is characterized by the increased production of reactive oxygen species (ROSs) and reactive nitrogen species (RNSs) [17], which are not sufficiently balanced by antioxidant defenses due to the disruption of redox signaling and adaptation. Knowledge of significant OS in stroke is important given its atherosclerotic or cardioembolic etiology [18,19,20,21]. OS promotes endothelial dysfunction, platelet activation, platelet–leukocyte aggregation and thrombus formation by the modification of fibrinogen [23,34,35]. ROSs include, for example, superoxide anion radical (O_2_^•−^), hydrogen peroxide, peroxyl radicals and hydroxyl radicals [36]. RNSs include nitric oxide (^•^NO) and peroxynitrite anion (ONOO^−^). The enzymatic oxidation of arginine produces NO by the three isoforms of NO synthases: inducible NO synthase (iNOS), endothelial NO synthase (eNOS) and neuronal NO synthase (nNOS). Isoforms iNOS and nNOS are harmful in ischemic strokes [37].

The presence of OS can be monitored using different methods, mainly LC-MS/MS, to measure carbonylated proteins, MDA, 4-hydroxy-2-nonenal and F2-isoprostanes [38]. MDA, belonging to the family of lipid–peroxidation biomarkers, is considered a standard OS marker. These biomarkers are also important triggers of stroke-related thrombosis [39]. Numerous clinical studies [40] found increased MDA levels in acute stroke patients’ plasma and saliva. In Cano et al. [40], 15 patients who were admitted with acute ischemic stroke were examined. Sampling was performed 24 h after stroke onset. MDA and NO levels were measured. The MDA levels significantly increased compared to healthy controls, whereas NO levels significantly decreased. Re et al. [41] analyzed the plasma activity of myeloperoxidase (MPO) and the concentrations of MDA and 4-hydroxynonenal before treatment in 50 consecutive stroke patients. The authors reported an increase in the levels of lipid peroxidation products and MPO activity. Considering the level of consciousness, the possible source of emboli, leukocyte count and relevant comorbid disease as covariates, only cardiac embolization was associated with this increase. In a study by Hajsl et al., it was also observed that the levels of plasmatic MDA were statistically significantly higher both in patients with significant carotid artery stenosis and acute ischemic stroke compared to the controls [22,42]. Moreover, in patients with acute coronary syndrome, it was revealed that concentrations of MDA tended toward higher values a few days after admission and gradually decreased during the follow-up period [43]. Rašić found a strong positive correlation between plaque scores and MDA in carotid arteries [44]. Enhanced concentrations of MDA created from lipoperoxides modify the apo B-100 protein part of otherwise oxidized LDL and make them even more atherogenic [45]. This is in agreement with our previous results, where we isolated LDL from plasma, modified them by either MDA or hypochloride and found markedly enhanced platelet adhesion on such materials [46]. It was proposed by Sigala that oxidized LDL in human carotid plaques is related to symptomatic carotid disease and lesion instability [16]. The authors also observed significantly increased MDA in atherosclerotic carotid plaques compared with normal lesions, but they found no significant differences in MDA concentrations between stable/unstable plaques and symptomatic/asymptomatic lesions. Nevertheless, this result could be substantially influenced by MDA measurement methods using colorimetric tests without the HPLC separation of colored products of 1-methyl-2-phenylindole with MDA, which can be formed from simple sugars [47]. Interestingly, patients with features of high risk-plaque as compared to stable plaques have often high serum levels of MDA, whereas non-significant differences in LDL levels, despite statin treatment [48]. Regarding these facts, MDA seems to be a very important factor involved in atherosclerotic plaque formation, with correlations relative to symptomatology. The precise correlation with CS has not been, however, previously studied.

In the present study, we performed a correlation between MDA levels and patient symptomatology. According to NASCET and ECST studies that remain the cornerstone of current guidelines, symptomatic patients were defined as having any symptoms (AF, TIA or BS) in the territory of CS within 6 months before the study. This concept was modified, after a thorough analysis of endarterectomy specimens by Verhoeven et al. [6], and the plaques of patients with AF were, then after, classified as stable. In view of the presented data, we decided to perform two correlations. The first correlation was based on the NASCET/ECST criteria of symptomatology, and the second is based on the knowledge that AF patients have the histological appearance of stable plaques. In the first correlation, statistical analysis showed that MDA plasma levels correlated with symptomatic plaques; however, the *p*-value did not reach the significance threshold (*p*-value = 0.232). In the second analysis, MDA plasma levels had the highest significant correlation of all studied parameters, reaching a *p*-value of 0.002. In such circumstances, MDA should be taken as a strong predictor of hemispheric brain strokes. Noteworthily, those results show that the pre-surgical evaluation of MDA in plasma has even higher predictive values than histological evaluations by AHA classifications.

## 5. Limitations of the Study

There was a limited number of patients, and MDA was the only marker of OS. Nevertheless, MDA is a reliable marker of OS, and there is a strong positive correlation between the plaque score and MDA in carotid arteries [44].

This study lacked any control cases. Thus, the clinical impact of the current results is limited and further studies including comparisons with other biomarkers of oxidative stress are needed.

## 6. Conclusions

Our study shows an analysis of endarterectomy specimens and the MDA plasma levels of 35 patients. Individual histological features of carotid plaque removed from the carotid artery were studied. Several histological features correlated with carotid symptomatology. As long as carotid plaques were classified based on the AHA classification, statistical significance reached the highest value among the histological parameters. The first correlation between asymptomatic and symptomatic (TIA, BS and AF) patients and MDA plasma levels was calculated to reach a *p*-value of 0.232. The second correlation was performed between the group of asymptomatic patients, including patients presented with AF and the hemispheric brain stroke group (BS and TIA). This correlation with MDA levels reached statistical significance, with a *p*-value of 0.002. According to our results, MDA seems to be a significant predictor of plaque vulnerability, which is even stronger than the AHA plaque classification. Thus, MDA is a valuable method for detecting unstable plaques and may impact the risk stratification of patients with CS and have an impact on decision-making processes when planning treatment strategies.

## Figures and Tables

**Figure 1 antioxidants-12-00506-f001:**
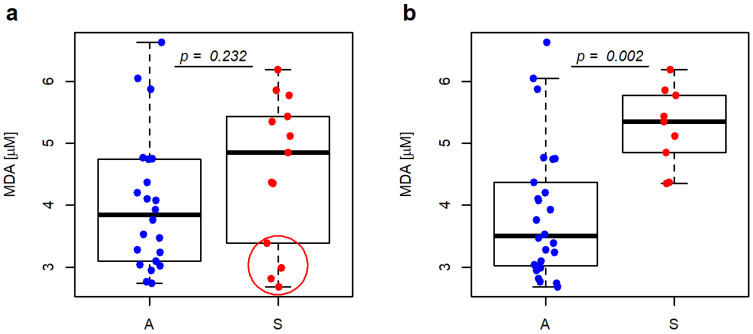
MDA levels are represented as boxplots—(**a**) asymptomatic (A—blue dots) and symptomatic (S—red dots) group, patients with AF (in red circle); (**b**) asymptomatic + AF (A) and symptomatic—hemispheric strokes only (S) group. Wilcoxon rank sum test with continuity correction. Vertical boxplots are constructed between the first (Q1) and third (Q3) quartile, with the horizontal median inside. The difference between Q3 and Q1 is called the interquartile range (IQR). Whiskers are drawn from Q1 and Q3 to minimal, respective maximal data points within the range of (Q1 − 1.5 IQR)–(Q3 + 1.5 IQR). Other data are outliers.

**Table 1 antioxidants-12-00506-t001:** Baseline characteristics. TAG—triglycerides; LDL—low-density lipoproteins; HDL—high-density lipoproteins; CKD-EPI—glomerular filtration. Variables are expressed as mean ± standard deviation.

	Asymptomatic (n = 22)	Symptomatic (n = 13)	*p*-Value
Sex (f/m)	4/18	5/8	0.185
Age (y)	69 ± 7.7	71 ± 6.9	0.561
Stenosis CTA (%)	76 ± 14	83 ± 10	0.435
Clinical characteristics			
Arterial hypertension	20 (91%)	10 (77%)	0.253
Diabetes mellitus	5 (23%)	5 (38%)	0.319
Smoker	13 (59%)	4 (31%)	0.105
Body mass index	29.16 ± 3.9	28.45 ± 3.8	0.596
Laboratory results			
Creatinine (µmol/L)	91.62 ± 15.43	84.5 ± 16.63	0.295
CKD-EPI (mL/min/1.73 m^2^)	75.43 ± 15.06	78.83 ± 16.85	0.640
Total cholesterol (mmol/L)	4.56 ± 1.20	4.27 ± 1.01	0.587
TAG (mmol/L)	1.95 ± 1.04	1.65 ± 0.47	0.549
LDL cholesterol (mmol/L)	2.51 ± 1.09	2.29 ± 0.99	0.922
HDL cholesterol (mmol/L)	1.15 ± 0.32	1.23 ± 0.30	0.525
Medical history			
History of myocardial infarction	5 (23%)	1 (8%)	0.254
History of stroke	5 (23%)	0 (0%)	0.063
Peripheral artery disease	2 (9%)	1 (8%)	0.593

**Table 2 antioxidants-12-00506-t002:** Histological features and AHA classification of carotid plaques based on the criteria of symptomatology by NASCET and ECST. *p*-value of statistical significance is highlighted.

Plaque Characteristics	Asymptomatic (n = 22)	Symptomatic (n = 13)	*p*-Value
Stable (AHA class IV,V,VIII)	14	2	**0.006**
Unstable (AHA class VI)	8	11	**0.006**
Eccentric	18	12	0.392
Concentric	4	1	0.392
Mainly atheroma	3	6	**0.033**
Mainly fibrous	9	1	**0.036**
Combined atheroma and fibrous	10	6	0.968
Without calcifications	4	2	0.832
Microcalcifications	8	9	0.060
Large calcifications	10	2	0.070
Myxoid changes	7	0	**0.023**
Hemorrhage—old	6	9	**0.015**
Hemorrhage—new	3	4	0.221
Without hemorrhage	13	3	**0.039**
Thrombosis above the plaque	0	1	0.187
Inflammation	19	12	0.593
Foam cells	11	6	0.826
Giant cells reaction	4	3	0.726
Hemosiderin/pigmentophage	12	6	0.631
Without neovascularisation	4	5	0.185
Small neovascularisation	9	7	0.458
Significant neovascularisation	9	1	**0.036**
Ossification	1	0	0.435

Bold in *p*-Value reflects the statistical significance *p* < 0.05.

**Table 3 antioxidants-12-00506-t003:** Histological features and AHA classification of carotid plaques. Asymptomatic + AF group contains patients presented with no symptoms or with AF within the previous 6 months. The hemispheric brain stroke group consists of patients who presented with TIA or brain stroke within the previous 6 months. *p*-value of statistical significance is highlighted.

Plaque Characteristics	Asymptomatic + AF (n = 26)	Hemispheric Brain Stroke (n = 9)	*p*-Value
Stable (AHA class IV,V,VIII)	15	1	**0.016**
Unstable (AHA class VI)	11	8	**0.016**
Eccentric	22	8	0.752
Concentric	4	1	0.752
Mainly atheroma	4	5	**0.017**
Mainly fibrous	10	0	**0.028**
Combined atheroma and fibrous	12	4	0.929
Without calcifications	5	1	0.577
Microcalcifications	11	6	0.208
Large calcifications	10	2	0.376
Myxoid changes	7	0	0.082
Hemorrhage—old	8	7	**0.014**
Hemorrhage—new	4	3	0.246
Without hemorrhage	15	1	**0.016**
Thrombosis above the plaque	1	0	0.551
Inflammation	22	9	0.211
Foam cells	13	4	0.774
Giant cells reaction	6	1	0.439
Hemosiderin/pigmentophage	15	3	0.208
Without neovascularisation	4	5	**0.017**
Small neovascularisation	13	3	0.387
Significant neovascularisation	9	1	0.179
Ossification	1	0	0.551

Bold in *p*-Value reflects the statistical significance *p* < 0.05.

## Data Availability

Data is contained within the article.

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
