# Peer review of "Oxidative Stress as a Reliable Biomarker of Carotid Plaque Instability: A Pilot Study"

_antioxidants, 2023, doi:10.3390/antiox12020506_

Round 1

Reviewer 1 Report

The manuscript entitled "Oxidative stress as a reliable biomarker of carotid plaque instability" brings  interesting information of the patients who have endarterectomy surgical procedure.

Still, several important observation have to be made:

Abstract

- line 16 - AHA classification - please define AHA abbreviation

- please define your groups more precisely in abstract. The method sections contains no useful information regarding your work.

Methods

- replace "carotid" with "carotid artery"

- line 34 - reformulate the sentence - is improper to write "burden their carriers with low risk" ; eventually you can replace with "patients with stable atherosclerotic plaque have a low risk"

- you wrote in two different places the aim of the study ; please write it once at the end of Introduction

- in Introduction you need to offer more information about the role of oxidative/nitrosative stress in atherosclerosis, and about the role of these parameters in diagnosis of complicated plaque.

Overall the Introduction chapter does not offer enough information about the subject of the manuscript!

Material and Methods

The Ethic Committee Approval is misssing.  Where the study was made? Please offer information about the hospital were you made the study. Also about the period of time you considered to enroll the patients. The inclusion/exclusion criteria are also missing. What kind of study did you make. Is a prospective/retrospective study or other category?

- line 53 -The groups you made are difficult to understand. - patients with symptoms more than 6 m were included in asymptomatic group ? Please reformulate the groups, because is very confusing.

Results

The table numbers are confusing. Some tables have no numbers. Please revise it.. Please write the number of the tables in the bottom and also the table's title. 

What do you mean by TAG ? It is about tryglicerides ? You wrote tryglicerols. Please write tryglycerides.

What kind of stroke had your patients? Ischemic/hemorrhagic? You need to be more precise. Same for myocardial infarction. 

Discussions 

All the discussions contains mixed text (some information you should write in introduction). The discussion chapter should contain only your results compared with other studies results.

Conclusions

Please write more details about the relevance of your results.

Reviewer 2 Report

The presented paper is a research study on the potential associations between plasma levels of MDA and features of vulnerable plaque and symptomatic internal carotid artery stenosis (ICAS). The study issue has no features of novelty, and so references that are out of date. I feel that in the form it is now, it is not suitable for publication. 

The study comprised 22 symptomatic and 13 asymptomatic patients who obtained endarterectomy for ICAS.

My major concerns are small study groups, insufficient description of symptomatic patients. The symptomatic ICAS should be asured from consulting neurologist, and addressed more carefully in methods.

The Authors seem not sure whether amaurosis fugax (AF) is associated with symptomatic ICAS, or asymptomatic ICAS. Please address this issue with consulting neurologist. I feel concerned that when AF was included by the Authors, the statistical tests did not obtained statistical significance, when AF was excluded from analysis the tests were significant. 

Table 1 - there should be given a degree of ICAS stenosis for each group. And it should be presented adequately to the Authors decision on the AF. 

Table 2 on the plaque histopathology should be presented as comparison of plaque features in symptomatic and asymptomatic ICAS with an inclusion of p-value. As it is presented this way, it has no value.

Discussion should address the issue of oxidative stress in the context of the Authors findings, not a review. Preferably the up-to-date findings

References should be upgraded

Reviewer 3 Report

The manuscript by Svoboda et al., investigate oxidative stress biomarker in patients with hemispheric brain stroke. Oxidative stress is a very well-known process in ischemic stroke. Therefore, this study is not novel, and the authors should have discussed their novelty. My detailed comments are below-

1. Information about oxidative stress and brain stroke could be more extensive in the introduction.

2. Author provided a lot of tabular data; while reviewing this manuscript, it took much work to follow all numerical without detailed information in figure legends. Also, it took a long time to follow table 2, because the heading says “Asymptomatic” and the legend says “Symptomatic”.

3. The sample number could be higher; therefore, the authenticity of the hypothesis is questionable without other OS markers like ELISA of enzymes, protein oxidation level, MPO, etc.

4. The presented figure’s quality is very poor; the author needs to improve before publication.

5. Discussion section is literally another “Introduction”, only a tiny paragraph discussing the MDA level. I will suggest rewriting this part after they perform additional OS markers. Otherwise, all text looks irrelevant.

6. I will suggest a detailed statistical analysis for MDA, like giving the all quartile (Q1, Q3), median value.

Round 2

Reviewer 1 Report

The revised manuscript is better nor, still you need to conduct further studies with more than one oxidative parameters. Eventually you need a control group for comparision (which you need to mention in the limitations of this study). taking in account all of these you should mention in the title this is a pilot study.

Reviewer 2 Report

Please, See attached file

Reviewer 3 Report

Although authors have improved writing  and fluency of this manuscript, but scintific soundness is same. I totally disagree, that because this is a communication one experiment is enough. I will suggest atleast one more experiment to cross check hypothesis.  
